# *Delftia acidovorans* secretes substances that inhibit the growth of *Staphylococcus epidermidis* through TCA cycle-triggered ROS production

**Tomotaka Ohkubo**[1,2], **Yasuhiko Matsumoto**[1], **Otomi Cho**[1], **Yuki Ogasawara**[2], **Takashi Sugita**[1] *

1 Department of Microbiology, Meiji Pharmaceutical University, Kiyose, Tokyo, Japan, 2 Department of Analytical Biochemistry, Meiji Pharmaceutical University, Kiyose, Tokyo, Japan

* sugita@my-pharm.ac.jp

**Data Availability Statement:** All data identified in the present study are provided in the paper and its Supplementary information (S1 Dataset). The RNA

## Abstract

The proportion of *Staphylococcus aureus* in the skin microbiome is associated with the severity of inflammation in the skin disease atopic dermatitis. *Staphylococcus epidermidis*, a commensal skin bacterium, inhibits the growth of *S. aureus* in the skin. Therefore, the balance between *S. epidermidis* and *S. aureus* in the skin microbiome is important for maintaining healthy skin. In the present study, we demonstrated that the heat-treated culture supernatant of *Delftia acidovorans*, a member of the skin microbiome, inhibits the growth of *S. epidermidis*, but not that of *S. aureus*. Comprehensive gene expression analysis by RNA sequencing revealed that culture supernatant of *D. acidovorans* increased the expression of genes related to glycolysis and the tricarboxylic acid cycle (TCA) cycle in *S. epidermidis*. Malonate, an inhibitor of succinate dehydrogenase in the TCA cycle, suppressed the inhibitory effect of the heat-treated culture supernatant of *D. acidovorans* on the growth of *S. epidermidis*. Reactive oxygen species production in *S. epidermidis* was induced by the heat-treated culture supernatant of *D. acidovorans* and suppressed by malonate. Further, the inhibitory effect of the heat-treated culture supernatant of *D. acidovorans* on the growth of *S. epidermidis* was suppressed by *N*-acetyl-L-cysteine, a free radical scavenger. These findings suggest that heat-resistant substances secreted by *D. acidovorans* inhibit the growth of *S. epidermidis* by inducing the production of reactive oxygen species via the TCA cycle.

## Introduction

Atopic dermatitis (AD) is a chronic inflammatory skin disease characterized by variable clinical features, such as relapsing pruritus and eczema [1,2]. Genetic and environmental factors are associated with the onset and exacerbation of AD [3–5]. The skin of AD patients exhibits an elevated pH and barrier disruption. Moreover, the skin microbial composition is altered in AD patients compared with healthy humans. Microbial diversity on the human skin contributes to maintain healthy skin by modulating immune responses [6–8]. In the skin of AD

seq data have been deposited at DDBJ/ENA/
GenBank: Submission ID = mpu_microbiology-
0001, BioProject = PRJDB11464, BioSample =
SAMD00294166, SAMD00294165, Accession
number = DRA011812.

**Funding:** This study was supported by the Japan
Society for the Promotion of Science, JSPS,
https://www.jsps.go.jp/english/index.html, (grant
number JP20K07208 to O.C., JP16K08384, and
JP19K07176 to T.S.). The funders did not play any
role in the study design, data collection and
analysis, decision to publish, or preparation of the
manuscript.

**Competing interests:** The authors have declared
that no competing interests exist.

patients, the microbial diversity is reduced due to an increased proportion of *Staphylococcus aureus* in the skin microbiome [4]. *S. aureus* exacerbates AD and induces Th2 cytokines and proteases in human skin [4,9,10]. On the other hand, *Staphylococcus epidermidis*, a coagulase-negative *Staphylococcus*, inhibits *S. aureus* growth by producing antimicrobial peptides and short-chain fatty acids [11,12]. The proportion of *S. aureus* in the skin of AD patients was decreased after transplantation of coagulase-negative *Staphylococcus* strains having antimicrobial activity [11]. Disrupting the balance between *S. aureus* and coagulase-negative *Staphylococcus* with antimicrobial activity against *S. aureus* in the skin microbiome is associated with exacerbation of AD [11,12]. The relative abundance of *S. aureus* is increased in the skin of patients with psoriasis (PS) as well as in those with AD [13,14]. *S. epidermidis* also inhibits the growth of *Cutibacterium acnes*, which is associated with acne vulgaris (AV) [15–17]. Therefore, *S. epidermidis* might play a key role in various skin disorders.

*Delftia acidovorans*, a gram-negative bacterium, is detected in the skin of AD patients by microbiome analysis [18–22]. *D. acidovorans* secretes a compound with antimicrobial activity against *S. aureus*, *Enterococcus faecalis*, *Acinetobacter baumannii*, *Klebsiella pneumoniae*, and *Pseudomonas aeruginosa* [23]. Thus, *D. acidovorans* may affect bacterial growth in the skin microbiome, but the effect of *D. acidovorans* on the growth of *S. epidermidis* remains unclear.

In the present study, we investigated the effect of *D. acidovorans* on the growth of *S. epidermidis* and elucidated its mechanisms. We found that the heat-treated culture supernatant (CS) of *D. acidovorans* inhibited *S. epidermidis* growth. Furthermore, we demonstrated that *D. acidovorans* heat-treated CS induced the production of reactive oxygen species (ROS) via the tricarboxylic acid cycle (TCA) cycle in *S. epidermidis*. Our findings provide important insight into how *D. acidovorans* affects the skin microbiome by inhibiting the growth of *S. epidermidis*, resulting in a skin microbiome imbalance related to various skin disorders.

## Materials & methods

### Reagents

Malonic acid, *N*-acetyl-L-cysteine (NAC), and gentamicin sulfate were purchased from Wako Pure Chemical Corporation (Osaka, Japan). Menadione was purchased from MilliporeSigma (St. Louis, MO, USA). 2,7-Dichlorodihydrofluorescein diacetate was purchased from Cayman Chemical (Ann Arbor, MI, USA).

### Strain and growth conditions

The NBRC100911 strain of *Staphylococcus epidermidis* (*S. epidermidis*) and the NBRC100910 strain of *Staphylococcus aureus* (*S. aureus*) were obtained from the National Institute of Technology and Evaluation (Tokyo, Japan). *S. epidermidis* and *S. aureus* strains were spread on nutrient agar (5 g/L sodium chloride [Wako Pure Chemical Corporation], 5 g/L beef extract [Becton Dickinson, Franklin Lakes, NJ, USA], 10 g/L hipolypepton [Wako Pure Chemical Corporation], and 15 g/L agar powder [Wako Pure Chemical Corporation] before autoclaving) and grown overnight at 37˚C. The JCM6218 strain of *D. acidovorans* was obtained from the Japan Collection of Microorganisms (Ibaraki, Japan). The *D. acidovorans* JCM6218 strain was spread on nutrient agar and grown overnight at 27˚C.

### *D. acidovorans* culture supernatant preparation

A colony of the *D. acidovorans* JCM6218 strain was inoculated in 3 mL nutrient broth (NB) and incubated overnight at 32˚C. The bacterial culture (3 mL) was inoculated into 300 mL NB and incubated overnight at 32˚C with shaking (200 rpm, ZWY-240 Incubator Shaker,

LABWIT Scientific, VIC, Australia). Bacterial cells were removed by centrifugation at 6300 rpm for 5 min (TOMY-MX305, TOMY Digital Biology Co. Ltd, Tokyo, Japan). The *D. acido-vorans* CS was filtered using the Vacuum Filtration 500 rapid filter MAX (TPP, Schaffhausen, Switzerland). *D. acidovorans* CS incubated for 30 min at 100˚C was used as *D. acidovorans* heat-treated CS in this study.

### Bacterial growth inhibition assay

Bacterial growth inhibition assays were performed according to a previous report [24]. Samples including *D. acidovorans* CS and inhibitors were diluted with NB to the appropriate concentrations and dispensed in 50-μL aliquots into a 96-well plate (TPP). *S. epidermidis* and *S. aureus* suspensions (2 x $10^4$ cells/ml) were prepared with NB and 50 μL was added to each well. After incubating at 37˚C for 12 h or 32˚C for 18 h, absorbance at 630 nm was measured using a microplate reader (iMark™ microplate reader; Bio-Rad Laboratories Inc., Hercules, CA, USA).

### RNA sequencing analysis

RNA-sequencing (RNA-seq) analysis was performed according to a previous report [24]. An *S. epidermidis* suspension was prepared with NB to an absorbance of 1 at 630 nm. The suspension was diluted in the same volume of *D. acidovorans* CS and incubated at 37˚C for 4 h. After incubation, RNA was extracted from the cells using EZ-Beads (AMR, Inc., Gifu, Japan) and a High Pure RNA Isolation kit (Roche, Basel, Switzerland) according to the manufacturer's instructions. The RNA-seq library was prepared from the RNA using an NEBNext rRNA Depletion Kit and NEBNext Ultra II RNA Library Prep Kit for Illumina (New England Biolabs Japan Inc., Tokyo, Japan) and subjected to RNA-seq analysis using MiSeq (Illumina Inc., San Diego, CA, USA). Gene function information was examined using the GenBank (https://www.ncbi.nlm.nih.gov/genbank/) and pathway information was analyzed using the Kyoto Encyclopedia of Genes and Genomes (KEGG; https://www.genome.jp/kegg/kegg_ja.html).

### Time-kill assay

The *S. epidermidis* suspension (1×$10^5$ cells/ml) was prepared with phosphate-buffered saline (PBS) and treated with 20% heat-treated NB, 20% *D. acidovorans* heat-treated CS, or gentamicin (final concentration 10 μg/mL). Each sample was incubated at 37˚C or 32˚C. Aliquots were serially diluted in PBS and 100 μL of the diluted samples was spread on a nutrient agar plate at 0, 3, and 6 h. The viable cell number was determined by counting the colonies on the plate after incubation for 24 h at 37˚C.

### Measuring ROS production

Quantification of ROS production in *S. epidermidis* with 2',7'-dichlorodihydrofluorescein diacetate) was performed according to previous reports [25,26]. *S. epidermidis* was suspended in 3 mL NB and incubated overnight at 37˚C with shaking (150 rpm, ZWY-240 incubator shaker, LABWIT Scientific). The culture was suspended in NB to an absorbance of 0.001 at 630 nm and exposed to 100 μM 2',7'-dichlorodihydrofluorescein diacetate (final concentration) for 1 h at 37˚C. After incubation, the samples were centrifuged at 14,000 rpm for 5 min (TOMY-MX105) and the supernatants were removed. The bacterial pellets were suspended in fresh NB and treated with *D. acidovorans* heat-treated CS or menadione, an ROS inducer. Fluorescence (λ excitation = 485 nm, λ emission = 538 nm) was measured using the Fluoroskan Ascent™ (Thermo Fisher Scientific, Waltham, MA, USA). Fluorescence was determined by subtracting the background fluorescence.

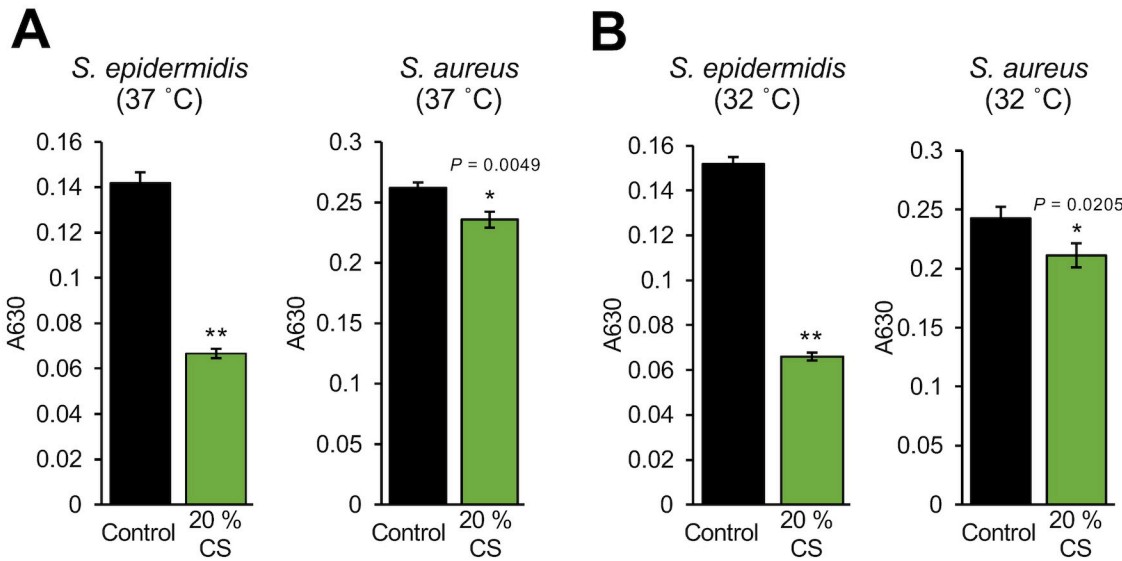

**Fig 1. Inhibitory activity of the culture supernatant of *D. acidovorans* against *S. epidermidis* growth.** *S. epidermidis* ($1\times10^4$ cells/mL) and *S. aureus* ($1\times10^4$ cells/mL) in NB were mixed with 20% *D. acidovorans* CS. After incubating at 37˚C for 12 h (A) or 32˚C for 18 h (B), absorbance at 630 nm was measured using a microplate reader. Error bars indicate the standard deviations (SD) of the means (n = 3). Statistical differences between groups were analyzed by the Student t-test. (*: $P < 0.05$, **: $P < 0.001$).

## Statistical analysis

Statistical differences between groups were analyzed by the Student *t*-test, Tukey-Kramer test, Dunnett test, and Williams test. All experiments were performed at least twice. Each experiment was performed in triplicate and error bars indicate the standard deviations of the means. A *P* value of less than 0.05 was considered statistically significant.

## Results

### Growth inhibitory effect of substances secreted by *D. acidovorans* against *S. epidermidis*

We investigated whether *D. acidovorans* produces substances that affect the growth of *S. epidermidis*. Compared with *S. aureus*, the growth of *S. epidermidis* at 37˚C or 32˚C was significantly inhibited by *D. acidovorans* CS (Fig 1). *D. acidovorans* heat-treated CS also inhibited *S. epidermidis* growth in a dose-dependent manner (Fig 2). We next examined whether the inhibitory activity of *D. acidovorans* heat-treated CS against *S. epidermidis* is bactericidal or bacteriostatic. *D. acidovorans* heat-treated CS did not decrease the viable number of *S. epidermis* in PBS (Fig 3). On the other hand, gentamicin, which has bactericidal activity, decreased the number of viable bacteria within 3 h (Fig 3). These results suggest that heat-stable substances secreted by *D. acidovorans* exhibit bacteriostatic activity, but not bactericidal activity, against *S. epidermidis*.

### Role of the *S. epidermidis* TCA cycle in the inhibitory effect of heat-stable substances secreted by *D. acidovorans*

To understand the mechanism of action of the substances secreted by *D. acidovorans*, we analyzed gene expression in *S. epidermidis* in response to the *D. acidovorans* culture supernatant (CS). Expression of 100 genes in *S. epidermidis* was increased more than 2-fold by the *D.*

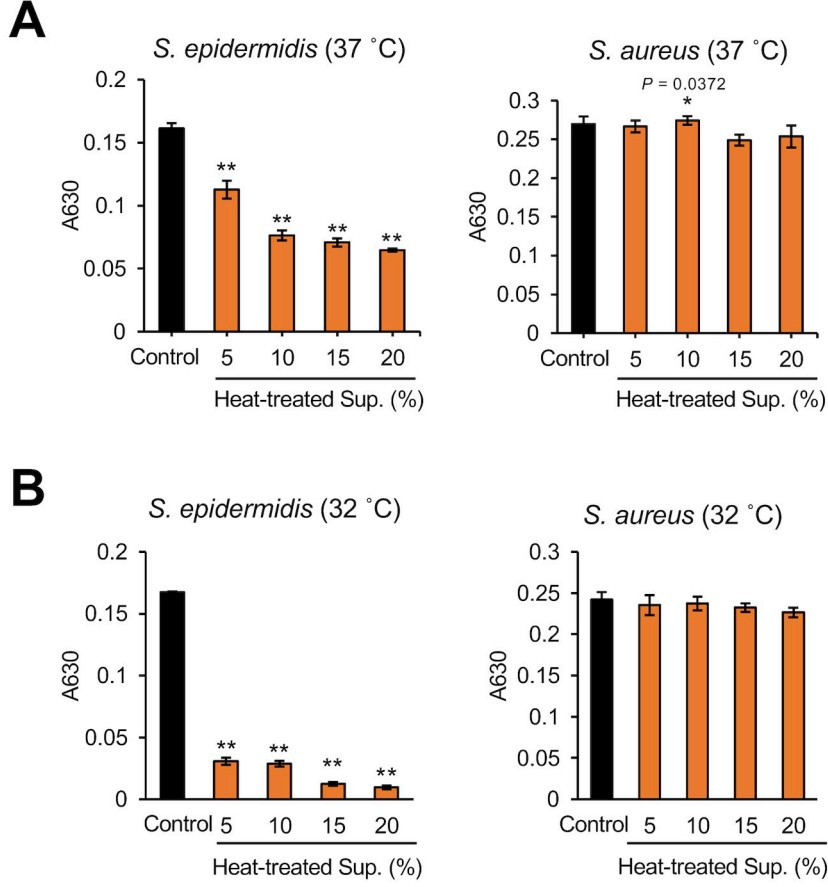

**Fig 2. Dose dependency of the inhibitory effect of heat-treated culture supernatant of *D. acidovorans* against *S. epidermidis* growth.** *S. epidermidis* ($1 \times 10^4$ cells/mL) and *S. aureus* ($1 \times 10^4$ cells/mL) in NB were mixed with 5% to 20% *D. acidovorans* heat-treated CS (Heat-treated CS). After incubating at 37˚C for 12 h (A) or 32˚C for 18 h (B), absorbance at 630 nm was measured using a microplate reader. Error bars indicate the SD of the means (n = 3). Statistical differences between groups were analyzed by the Williams test. ($^*$: $P < 0.05$, $^{**}$: $P < 0.001$).

*acidovorans* CS and expression of 7 genes was decreased to less than half (S1 and S2 Tables in S1 File). Gene ontology term analysis revealed that *S. epidermidis* genes related to glycolysis, the TCA cycle, and the response to oxidative stress were upregulated by the *D. acidovorans* CS (Fig 4A). Pathway analysis showed that expression of the 6 genes related to glycolysis and 3 genes related to the TCA cycle in *S. epidermidis* was increased more than 2-fold by the *D. acidovorans* CS (Fig 4B). Because pyruvate produced by glycolysis triggers TCA cycle activation, we focused on the relationship between activation of the TCA cycle, which is downstream of glycolysis, and the inhibitory activity of *D. acidovorans* heat-treated CS against *S. epidermidis*. Malonic acid (4 mM), a TCA cycle inhibitor, suppressed the inhibitory effect of the *D. acidovorans* heat-treated CS against *S. epidermidis* growth (Fig 4C). These results suggest that substances secreted by *D. acidovorans* inhibit the growth of *S. epidermidis* via TCA cycle regulation.

## Heat-stable substances secreted by *D. acidovorans* induce ROS production in *S. epidermidis*

The TCA cycle is linked to ROS production in *S. epidermidis* [27]. ROS production in *S. epidermidis* was induced by exposure to the *D. acidovorans* heat-treated CS (Fig 5A and 5B). The

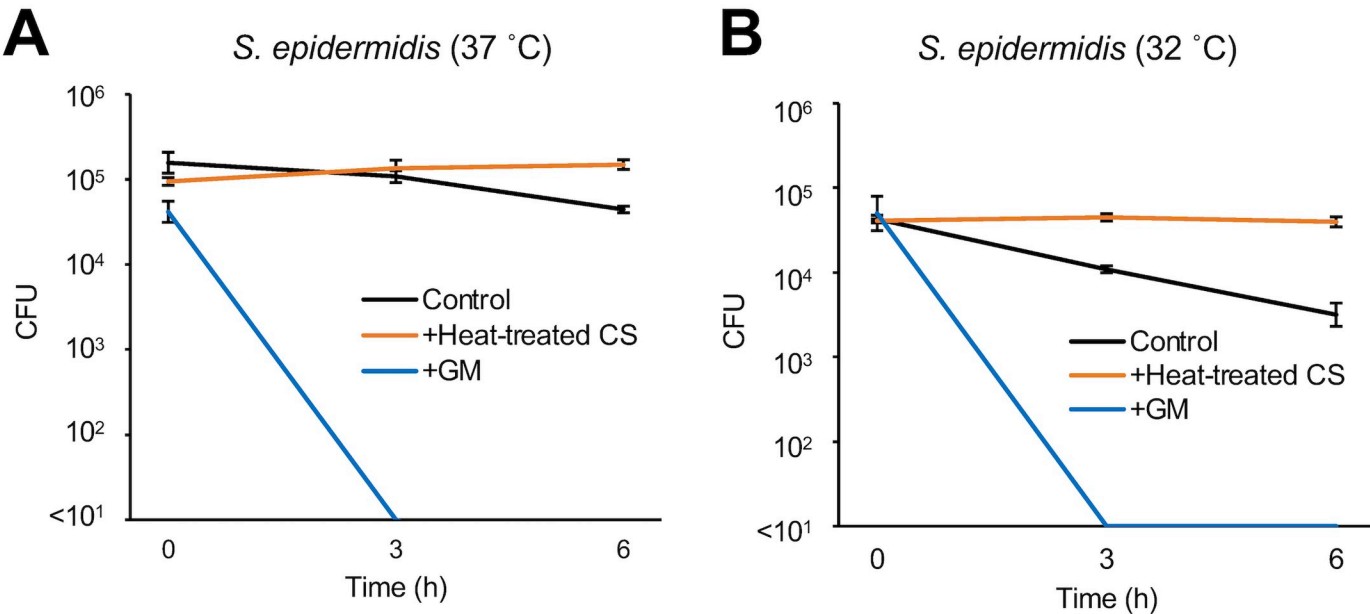

**Fig 3. Time-kill assay of heat-treated culture supernatant of *D. acidovorans* against *S. epidermidis*.** *S. epidermidis* ($1\times10^5$ cells/mL) in PBS was mixed with 20% NB (Control), 20% *D. acidovorans* heat-treated CS (Heat-treated CS), and 10 μg/mL gentamicin (GM) at 37˚C (A) or 32˚C (B). Each line shows the number of viable cells of *S. epidermidis* at 0, 3, and 6 h. Error bars indicate the SD of the means (n = 3).

addition of malonic acid inhibited the ROS production in *S. epidermidis* induced by the *D. acidovorans* heat-treated CS (Fig 5C and 5D). Addition of a free radical scavenger, NAC, suppressed the *S. epidermidis* growth inhibition in a dose-dependent manner (Fig 6). These results suggest that substances secreted by *D. acidovorans* inhibit the growth of *S. epidermidis* via TCA cycle-triggered ROS production (Fig 7).

## Discussion

The findings of the present study demonstrated that heat-stable substances secreted by *D. acidovorans* inhibited the growth of *S. epidermidis*, but not that of *S. aureus*. The heat-stable substances induced ROS production through the TCA cycle in *S. epidermidis*. Moreover, the inhibitory effect against *S. epidermidis* growth by the heat-stable substances was suppressed by NAC, a radical scavenger. Our findings suggest that *D. acidovorans* secretes heat-stable substances that inhibit *S. epidermidis* growth by TCA cycle-triggered ROS production.

A previous study demonstrated that *D. acidovorans* secreted heat-stable antimicrobial substances that inhibit the growth of *S. aureus* [23]. Therefore, we assessed the inhibitory activity of *D. acidovorans* heat-treated CS against *S. epidermidis*. We demonstrated that heat-stable substances secreted by *D. acidovorans* induced the expression of genes related to glycolysis and the TCA cycle, and the inhibitory effect against growth of *S. epidermidis* was suppressed by malonic acid, an inhibitor of succinate dehydrogenase in the TCA cycle. On the other hand, 4 mM malonic acid alone inhibited *S. epidermidis* growth, but this inhibitory activity was suppressed by the heat-stable substances of *D. acidovorans*. Therefore, we assumed that the inhibitory effect of malonic acid on the TCA cycle competed with the increased gene expression in the TCA cycle induced by the heat-stable substances secreted by *D. acidovorans*. The finding also suggests that the heat-stable substances of *D. acidovorans* affect the TCA cycle in *S. epidermidis*. We also assumed that the effect heat-stable substances of *D. acidovorans* on the TCA

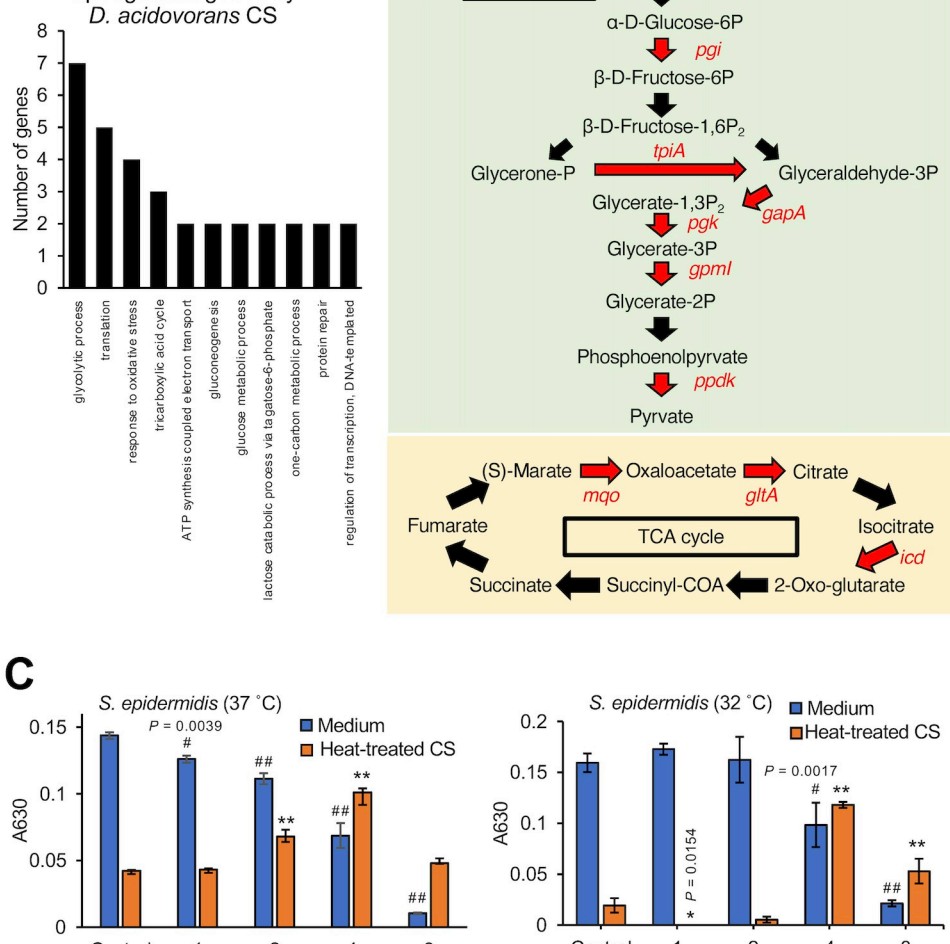

**Fig 4. Suppressive effect of a TCA cycle inhibitor on the inhibitory effect of heat-treated culture supernatant of *D. acidovorans* against *S. epidermidis* growth.** (A) Gene expression analysis of *S. epidermidis* ($1 \times 10^9$ cells/mL) mixed with 50% *D. acidovorans* CS at 37˚C for 4 h. The graphs show the number of upregulated genes in *S. epidermidis* after *D. acidovorans* CS treatment. (B) Upregulated genes related to glycolysis and the TCA cycle. Red letters represent genes that increased more than 2-fold after treatment with *D. acidovorans* CS in *S. epidermidis*. (C) *S. epidermidis* ($1 \times 10^4$ cells/mL) in NB was mixed with 1–4 mM malonic acid and 15% *D. acidovorans* heat-treated CS (Heat-treated CS). After incubating at 37˚C for 12 h or 32˚C for 18 h, absorbance at 630 nm was measured using a microplate reader. Error bars indicate the SD of the means (n = 3). Statistical differences of each group were analyzed by the Dunnett test. (#: $P < 0.05$ (Medium), ##: $P < 0.001$ (Medium), *: $P < 0.05$ (Heat-treated CS), **: $P < 0.001$ (Heat-treated CS)).

cycle of *S. epidermidis* led to an increase in the ROS production because the ROS production induced by the heat-stable substances of *D. acidovorans* was inhibited by malonic acid. Furthermore, the suppressive effect of NAC suggested that the induced ROS production in *S. epidermidis* caused the inhibitory effect of the heat-stable substances of *D. acidovorans* against the growth of *S. epidermidis*. *S. aureus*, on the other hand, produces staphyloxanthin, an antioxidant that confers resistance to ROS [28,29]. We speculated that the heat-stable substances of *D. acidovorans* exhibit no inhibitory effects against *S. aureus* growth because of differences in the resistance to ROS.

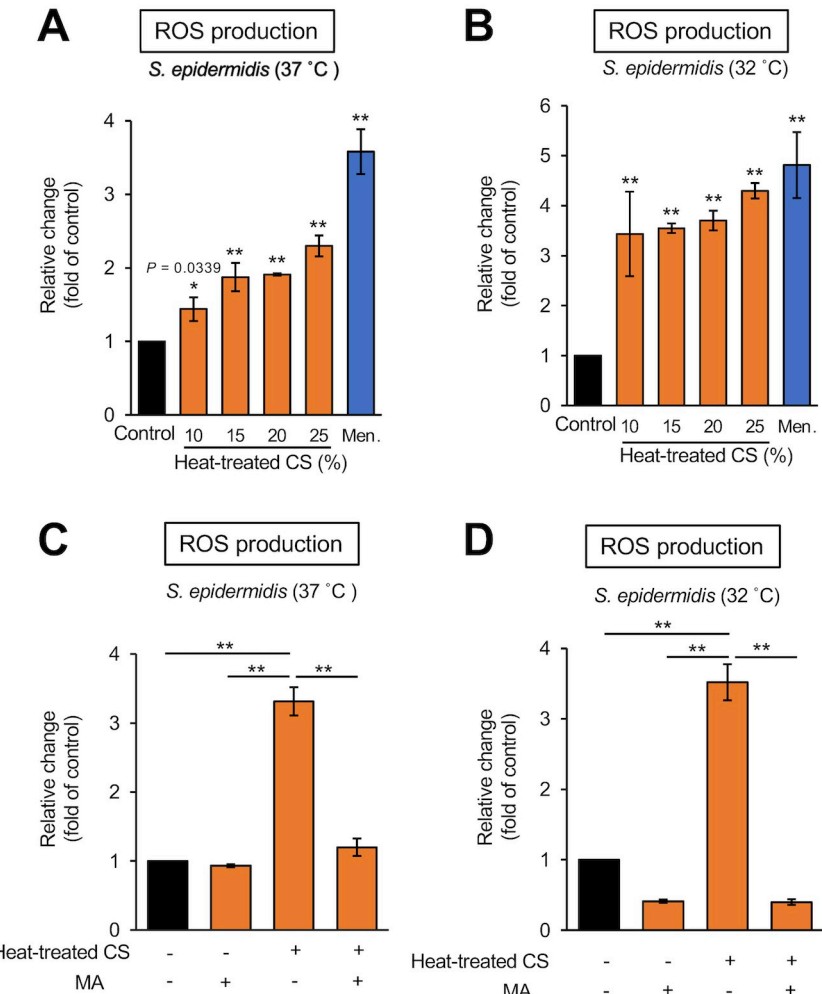

**Fig 5. ROS production induced in *S. epidermidis* by heat-treated culture supernatant of *D. acidovorans*.** (A, B) ROS production in *S. epidermidis* ($1 \times 10^6$ cells/mL) mixed with NB (Control), 10–25% *D. acidovorans* heat-treated CS (Heat-treated CS), or 70 μg/mL menadione (Men.) at 37˚C (A) or 32˚C (B) for 24 h. Menadione was used as a positive control. Fluorescence (λ excitation = 485 nm, λ emission = 538 nm) was measured using the Fluoroskan Ascent™. Error bars indicate the SD of the means (n = 3). Statistical differences between groups were analyzed by the Dunnett test. (*: $P < 0.05$, **: $P < 0.001$). (C, D) ROS production in *S. epidermidis* ($1 \times 10^6$ cells/mL) mixed with NB (Control), 4 mM malonic acid (MA), 25% *D. acidovorans* heat-treated CS (Heat-treated CS), or 4 mM MA + 25% heat-treated CS at 37˚C (C) or 32˚C (D) for 24 h. Fluorescence (λ excitation = 485 nm, λ emission = 538 nm) was measured using the Fluoroskan Ascent™. Error bars indicate the SD of the means (n = 3). Statistical differences between groups were analyzed by the Tukey-Kramer test. (*: $P < 0.05$, **: $P < 0.001$).

*D. acidovorans* is often isolated and detected by microbiome analysis from clinical samples, including from the skin of patients with AD [18,19,30–34]. We found that heat-stable substances secreted by *D. acidovorans* inhibited the growth of *S. epidermidis*, but not the growth of *S. aureus*. In the skin of patients with AD, the relative abundance of *S. aureus* is significantly increased, and enterotoxin or protease derived from *S. aureus* exacerbates inflammation of AD [4,9,10]. The relative abundance of *S. aureus* is also increased in the skin of patients with PS and Th17 cytokines are induced by *S. aureus* [13,14]. *S. epidermidis* inhibits the growth of *S. aureus*, which might be associated with AD and PS [11,12]. Moreover, *S. epidermidis* inhibits the growth of *C. acnes*, which is associated with AV [15–17]. We therefore hypothesized that

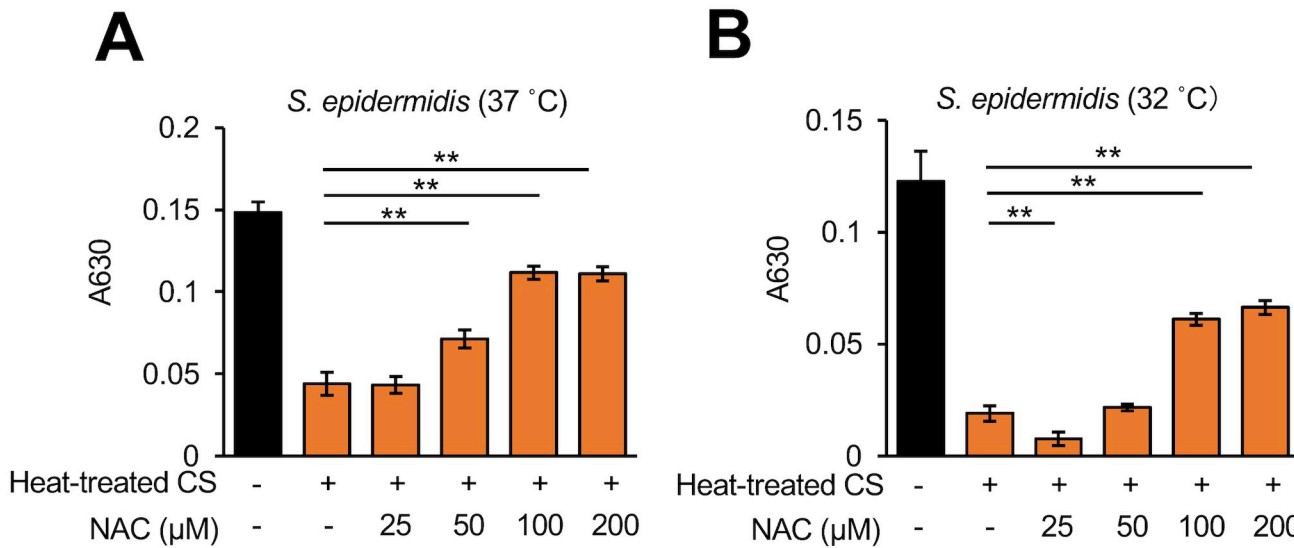

**Fig 6. Suppressive effect of *N*-acetyl-L-cysteine on the inhibitory activity of heat-treated culture supernatant of *D. acidovorans* against *S. epidermidis* growth.** *S. epidermidis* ($1\times10^4$ cells/mL) in NB was mixed simultaneously with 15% *D. acidovorans* heat-treated CS (Heat-treated CS) and 25–200 μM *N*-acetyl-cysteine (NAC). After incubating at 37°C for 12 h (A) or 32°C for 18 h (B), absorbance at 630 nm was measured using a microplate reader. Error bars indicate the SD of the means (n = 3). Statistical differences between groups were analyzed by the Tukey-Kramer test. (*: $P < 0.05$, **: $P < 0.001$).

the inhibitory activity of substances secreted by *D. acidovorans* against *S. epidermidis* is indirectly associated with exacerbation of AD, PS, and AV. *D. acidovorans* might be related to various skin disorders. Additional studies are needed to elucidate the relationship between *D. acidovorans* and AD, PS, and AV.

Previous studies reported that *D. acidovorans* secretes delftibactin, which has antimicrobial activity against *S. aureus* [23,35]. Delftibactin reacts in a solution containing $AuCl_3$ and forms a gold precipitation, resulting in detoxification by chelating $Au^{3+}$ [35]. We demonstrated that the *D. acidovorans* heat-treated CS used in this study did not form a gold precipitation by mixing it with a solution containing $AuCl_3$ (S1 Fig in S1 File). Furthermore, gold toxicity against *S. epidermidis* was not inhibited by *D. acidovorans* heat-treated CS (S2 Fig in S1 File). Therefore, we assumed that the *D. acidovorans* heat-treated CS used in this study may not contain a sufficient amount of delftibactin to inhibit the growth of *S. aureus*. We considered that differences in both the strain and culture conditions between previous studies and the present study could affect the composition of delftibactin. Tejman-Yarden *et al.* indicated the existence of a fraction other than the delftibactin fraction that exerts antimicrobial activity in the culture supernatant of *D. acidovorans*. In addition, a substance characterized as $C_{39}H_{68}N_{14}O_{17}$, which is similar to the composition of delftibactin is present in the delftibactin fraction [23]. It is possible that the active substances focused on in this study are in these fractions. Identifying the active substances in the *D. acidovorans* heat-treated CS is an important topic for future studies.

## Conclusion

*D. acidovorans* secretes heat-stable substances that have inhibitory activity against *S. epidermidis* growth through TCA cycle-triggered-ROS production. How *D. acidovorans* affects the proportion of *S. aureus* and *S. epidermidis* in the human skin microbiome is an important topic for future studies.

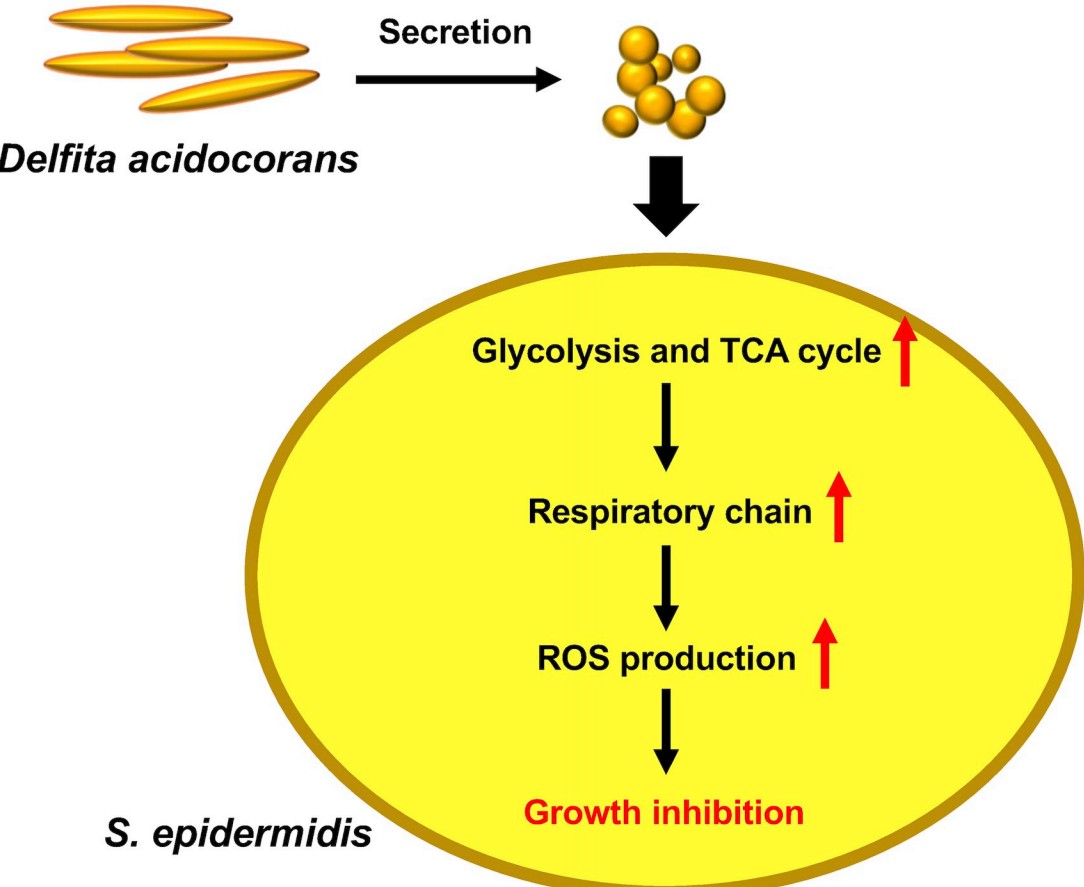

**Fig 7. Model of the action of heat-stable substances secreted by *D. acidovorans* against *S. epidermidis*.** Heat-stable substances secreted by *D. acidovorans* induce the expression of the genes related to glycolysis and the TCA cycle and subsequently increase ROS production in *S. epidermidis*. The induced ROS production leads to growth inhibition in *S. epidermidis*.

## Supporting information

**S1 File.**
(DOCX)

**S1 Dataset.**
(XLSX)

## Author Contributions

**Conceptualization:** Yasuhiko Matsumoto, Takashi Sugita.

**Data curation:** Tomotaka Ohkubo, Otomi Cho.

**Formal analysis:** Tomotaka Ohkubo, Otomi Cho.

**Funding acquisition:** Otomi Cho, Takashi Sugita.

**Investigation:** Tomotaka Ohkubo, Otomi Cho.

**Project administration:** Yasuhiko Matsumoto, Yuki Ogasawara, Takashi Sugita.

**Resources:** Takashi Sugita.

**Supervision:** Yasuhiko Matsumoto, Takashi Sugita.

**Validation:** Tomotaka Ohkubo, Otomi Cho.

**Visualization:** Tomotaka Ohkubo, Yasuhiko Matsumoto.

**Writing – original draft:** Tomotaka Ohkubo.

**Writing – review & editing:** Yasuhiko Matsumoto, Yuki Ogasawara, Takashi Sugita.

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
