## [Decision Letter · Decision Letter 0]

25 Mar 2021

PONE-D-21-01227

Delftia acidovorans secretes substances that inhibit the growth of Staphylococcus epidermidis through TCA cycle-triggered ROS production

PLOS ONE

Dear Dr. Sugita,

Thank you for submitting your manuscript to PLOS ONE. After careful consideration, we feel that it has merit but does not fully meet PLOS ONE’s publication criteria as it currently stands. Therefore, we invite you to submit a revised version of the manuscript that addresses the points raised during the review process.

We look forward to receiving your revised manuscript.

Kind regards,

Rashid Nazir

Academic Editor

PLOS ONE

Journal Requirements:

Additional Editor Comments (if provided):

Please revise your manuscript according to the reviewers feedback.

Reviewers' comments:

Reviewer's Responses to Questions

**Comments to the Author**

1. Is the manuscript technically sound, and do the data support the conclusions?

Reviewer #1: Partly

Reviewer #2: Yes

2. Has the statistical analysis been performed appropriately and rigorously? 

Reviewer #1: Yes

Reviewer #2: Yes

3. Have the authors made all data underlying the findings in their manuscript fully available?

Reviewer #1: No

Reviewer #2: No

4. Is the manuscript presented in an intelligible fashion and written in standard English?

Reviewer #1: No

Reviewer #2: Yes

5. Review Comments to the Author

Reviewer #1: Dear authors,

You mentioned atopic dermatitis in your manuscript. However, the relation between your findings and atopic dermatitis is unclear. For example, in other inflammatory skin disorders, such as psoriasis, the skin microbium will be changed as well. in other words, even though your findings may also be applicable to other skin disorders, there is no such discussion in your manuscript and/or the discussion of the relevance and significance of your findings with regards to atopic dermatitis is lacking. So please clarify your hypothesis and provide a discussion of the results and importance of your findings in atopic dermatitis.

English needs to be edited.

Reviewer #2: Rather than mentioning your findings in the end of introduction, why don’t you add the work-hypothesis or objectives? It will thus help the student-readers to have an overview of a typical research article. Moreover, the introduction should briefly be extended to better comprehend the study topic.

Does 100oC heat treatment (for 30 min) to bacterial culture supernatant is ok for metabolites integrity? What about the stability of the chemicals? I would appreciate if the authors may provide the rationale and appropriate literature references for this temperature treatment. And, what were the heat stable substance in this treated culture?

Why D. acidovorans JCM6218 strain was grown at 27℃ and subsequently on 32℃, if this bacterium is usually present on skin surfaces?

Line-153 says that D. acidovorans heat-treated CS did not decrease the viable number of S. epidermis (Fig 3); while Line 155-6 say that heat-stable substances secreted by D.

acidovorans have bacteriostatic activity against S. epidermidis. Please clarify the potential contradiction.

As per your data, 6-genes related to glycolysis and 3 genes related to the TCA cycle in S. epidermidis was increased more than 2-fold by the D. acidovorans CS; and then malonic

acid (4 mM) was used to verify the inhibitory effect via TCA cycle. I wonder why authors were convinced to verify their 3-genes TCA cycle and not 6-genes glycolysis? Please describe your rationale for this selection.

Why figure 7 data is presented in discussion and not in results section of the ms? Moreover, I suggest authors to further infer their results to furnish their discussion with more literature studies. For instance, authors could think of chemical substances other than delftibactin only which may inhibit the target bacterium in this case.

At present, I have the feeling that authors have just briefly reported their work; they could rather better describe their findings with sufficient discussion points and literature review in the introduction.

Figure legends are within the ms-text (without figures which are rightly in the end) and sometimes in interrogative case. I would suggest to keep the legends narrative, and all legends together after references.

RNA seq data should be available on some public database and be cited in ms accordingly.

6. PLOS authors have the option to publish the peer review history of their article (what does this mean?). If published, this will include your full peer review and any attached files.

Reviewer #1: **Yes: **Azin Ayatollahi, MD

Reviewer #2: **Yes: **Rashid Nazir

---

## [Author Response · Author response to Decision Letter 0]

29 Apr 2021

Response to reviewer’s comments

1. Is the manuscript technically sound, and do the data support the conclusions?

Reviewer #1: Partly

Reviewer #2: Yes

We now state in the revised manuscript that all experiments were replicated at least twice, as follows: "All experiments were performed at least twice." (Materials & Methods section, page 10, line 148)

The sample sizes for each experiment are provided in the figure legends of the revised manuscript (pages 24-27, lines 406-472).

2. Has the statistical analysis been performed appropriately and rigorously?

Reviewer #1: Yes

Reviewer #2: Yes

Thank you very much for reviewing our manuscript.

3. Have the authors made all data underlying the findings in their manuscript fully available?

Reviewer #1: No

Reviewer #2: No

We added the S1 Dataset file, which includes raw data related to our results, to the supplementary information of the revised manuscript (page 17, lines 262-264). Moreover, our data related to the RNA sequence analysis have been deposited at DDBJ/ENA/GenBank: Submission ID = mpu_microbiology-0001, BioProject = PRJDB11464 BioSample =SAMD00294166, SAMD00294165, Accession number = DRA011812. A data availability statement including this information has been added to the revised manuscript (page 17, lines 262-267).

4. Is the manuscript presented in an intelligible fashion and written in standard English?

Reviewer #1: No

Reviewer #2: Yes

Following the referee’s comment, the revised manuscript was edited by professional native English-speaking science editors (SciTechEdit International, LLC, CO, USA).

5. Review Comments to the Author

Reviewer #1: Dear authors,

You mentioned atopic dermatitis in your manuscript. However, the relation between your findings and atopic dermatitis is unclear. For example, in other inflammatory skin disorders, such as psoriasis, the skin microbiome will be changed as well. in other words, even though your findings may also be applicable to other skin disorders, there is no such discussion in your manuscript and/or the discussion of the relevance and significance of your findings with regards to atopic dermatitis is lacking. So please clarify your hypothesis and provide a discussion of the results and importance of your findings in atopic dermatitis. English needs to be edited.

According to the reviewer’s comment, we now mention the relationship between the skin microbiome and skin disorders such as psoriasis and acne vulgaris in the Introduction and Discussion sections of the revised manuscript (pages 4-5, lines 53-56; pages 15-16, lines 227-233).

Pages 4-5, lines 53-56.

"The relative abundance of S. aureus is increased in the skin of patients with psoriasis (PS) as well as in those with AD. S. epidermidis also inhibits the growth of Cutibacterium acnes, which is associated with acne vulgaris (AV). Therefore, S. epidermidis might play a key role in various skin disorders."

Pages 15-16, lines 227-233.

"In the skin of patients with AD, the relative abundance of S. aureus is significantly increased, and enterotoxin or protease derived from S. aureus exacerbates inflammation of AD. The relative abundance of S. aureus is also increased in the skin of patients with PS and Th17 cytokines are induced by S. aureus. S. epidermidis inhibits the growth of S. aureus, which might be associated with AD and PS. Moreover, S. epidermidis inhibits the growth of C. acnes, which is associated with AV."

We also added our hypothesis and provide a discussion of the results and importance of our findings in relation to AD in the revised manuscript (page 16, lines 233-237).

Page 16, lines 233-237.

“We therefore hypothesized that the inhibitory activity of substances secreted by D. acidovorans against S. epidermidis is indirectly associated with exacerbation of AD, PS, and AV. D. acidovorans might be related to various skin disorders. Additional studies are needed to elucidate the relationship between D. acidovorans and AD, PS, and AV."

Following the referee’s comment, the revised manuscript was edited by professional native English-speaking science editors (SciTechEdit International, LLC, CO, USA).

Reviewer #2: Rather than mentioning your findings in the end of introduction, why don’t you add the work-hypothesis or objectives? It will thus help the student-readers to have an overview of a typical research article. Moreover, the introduction should briefly be extended to better comprehend the study topic.

According to the reviewer’s comment, we added the following objective at the end of the Introduction section in the revised manuscript (page 5, lines 63-64).

Page 5 lines 63-64.

"In the present study, we investigated the effect of D. acidovorans on the growth of S. epidermidis and elucidated its mechanisms."

Does 100 ˚C heat treatment (for 30 min) to bacterial culture supernatant is ok for metabolites integrity?

The substances secreted by D. acidovorans may be heat-sensitive. On the other hand, we confirmed that the antimicrobial activity of substances secreted by D. acidovorans was not decreased by heat treatment at 100˚C for 30 min (Fig. 1 and Fig. 2). Therefore, we assumed that substances secreted by D. acidovorans are stable to 100˚C heat treatment for at least 30 min.

What about the stability of the chemicals? 

Because D. acidovorans CS exhibits antimicrobial activity after heat treatment at 100˚C, we assumed that substances secreted by D. acidovorans are heat-stable (Fig. 1 and Fig. 2).

In addition, we confirmed that the culture supernatant of D. acidovorans stored at 4˚C for 1 year exhibits the same amount of antimicrobial activity as a fresh sample. Therefore, we assumed that the substances were also stable in nutrient broth. 

I would appreciate if the authors may provide the rationale and appropriate literature references for this temperature treatment. And, what were the heat stable substance in this treated culture?

According to the reviewer’s comment, we added a sentence to the Discussion section of the revised manuscript (page 14, lines 203-205).

Page 14, lines 203-205.

“A previous study demonstrated that D. acidovorans secreted heat-stable antimicrobial substances that inhibit the growth of S. aureus (Tejman-Yarden N., et al., Front Microbiol. 10:2377, 2019). Therefore, we assessed the inhibitory activity of D. acidovorans heat-treated CS against S. epidermidis.“

The active substance has not yet been identified. We added the following sentence to the Discussion section of the revised manuscript (page 17, lines 252-254).

Page 17, lines 252-254.

“Identifying the active substances in the D. acidovorans heat-treated CS is an important topic for future studies.”

Why D. acidovorans JCM6218 strain was grown at 27℃ and subsequently on 32℃, if this bacterium is usually present on skin surfaces?

A previous study reported that D. acidovorans, a common environmental bacteria, can be grown at 4–41˚C (Wen A, et al., Int J Syst Bacteriol. 2:567-76, 1999). In addition, we confirmed that D. acidovorans JCM6218 can also be grown at 27˚C without any problem. Therefore, we incubated the D. acidovorans JCM6218 strain at 27˚C from stock before using it to perform the assay at 32˚C (temperature of human skin).

Reference：A Wen et al. Int J Syst Bacteriol. 2:567-76, 1999. doi: 10.1099/00207713-49-2-567.

Line-153 says that D. acidovorans heat-treated CS did not decrease the viable number of S. epidermis (Fig 3); while Line 155-6 say that heat-stable substances secreted by D. acidovorans have bacteriostatic activity against S. epidermidis. Please clarify the potential contradiction.

According to the reviewer’s comment, we added sentences about the bacteriostatic activity of the D. acidovorans CS in the Results section of the revised manuscript (page 11, lines 158-164). 

Chemicals such as antibiotics with bacteriostatic activity inhibit the growth of a bacterium, but do not kill cells. On the other hand, chemicals with bactericidal activity inhibit the growth of a bacterium and kill the cells.

We examined whether the D. acidovorans heat-treated CS had bactericidal activity against S. epidermidis in phosphate buffered saline (PBS). S. epidermidis cannot grow in PBS, but can survive at least 12 h. Gentamicin was used as a positive control because it exhibits bactericidal activity. The addition of gentamicin decreased the number of viable bacteria within 3 h, whereas the addition of D. acidovorans heat-treated CS had no effect (Fig 3). The D. acidovorans heat-treated CS, however, has the potential to inhibit the growth of S. epidermidis in a nutrient broth. Because the D. acidovorans heat-treated CS, which can inhibit the growth of S. epidermidis in a nutrient broth, did not kill the S. epidermidis cells, we assumed that the inhibitory activity of D. acidovorans heat-treated CS against S. epidermidis is bacteriostatic.

Page 11, lines 158-164.

“We next examined whether the inhibitory activity of D. acidovorans heat-treated CS against S. epidermidis is bactericidal or bacteriostatic. D. acidovorans heat-treated CS did not decrease the viable number of S. epidermis in PBS (Fig 3). On the other hand, gentamicin, which has bactericidal activity, decreased the number of viable bacteria within 3 h (Fig 3). These results suggest that heat-stable substances secreted by D. acidovorans exhibit bacteriostatic activity, but not bactericidal activity, against S. epidermidis.”

As per your data, 6-genes related to glycolysis and 3 genes related to the TCA cycle in S. epidermidis was increased more than 2-fold by the D. acidovorans CS ; and then malonic acid (4 mM) was used to verify the inhibitory effect via TCA cycle. I wonder why authors were convinced to verify their 3-genes TCA cycle and not 6-genes glycolysis? Please describe your rationale for this selection.

According to the reviewer’s comment, we added a sentence to the Results section of the revised manuscript (page 12, lines 176-179).

Because pyruvate produced by glycolysis triggers the TCA cycle, we speculated that induction of genes related to glycolysis leads to activation of the TCA cycle. In this study, genes of S. epidermidis related to both glycolysis and the TCA cycle were upregulated by D. acidovorans CS. Therefore, we focused on activation of the TCA cycle, which is downstream of glycolysis.

Page 12, lines 176-179.

"Because pyruvate produced by glycolysis triggers TCA cycle activation, we focused on the relationship between activation of TCA cycle, which is downstream of glycolysis, and the inhibitory activity of D. acidovorans heat-treated CS against S. epidermidis."

Why figure 7 data is presented in discussion and not in results section of the ms? 

According to the reviewer’s comment, we moved Figure 7 to the Results section in the revised manuscript (Page 13, line 192). 

Moreover, I suggest authors to further infer their results to furnish their discussion with more literature studies. For instance, authors could think of chemical substances other than delftibactin only which may inhibit the target bacterium in this case.

According to the reviewer’s suggestion, we added the following sentences to the Discussion section of the revised manuscript to describe that active substances besides delftibactin are possibly presented by D. acidovorans CS. 

"Tejman-Yarden et al. indicated the existence of a fraction other than the delfibactin fraction that exerts antimicrobial activity in the culture supernatant of D. acidovorans. In addition, a substance characterized as C39H68N14O17, which is similar to the composition of delftibactin, is present in the delfibactin fraction. It is possible that the active substances focused on in this study are in these fractions. Identifying the active substances in the D. acidovorans heat-treated CS is an important topic for future studies." (pages 16-17, lines 248-254)

At present, I have the feeling that authors have just briefly reported their work; they could rather better describe their findings with sufficient discussion points and literature review in the introduction.

According to the reviewer’s suggestion, we added sentences to the Introduction and Discussion sections of the revised manuscript (pages 4-5, lines 53-56; page 5, lines 63-64; page 5, lines 67-70; pages 15-16, lines 227-237).

Page 4-5, lines 53-56.

" The relative abundance of S. aureus is increased in the skin of patients with psoriasis (PS) as well as in those with AD (Totté JEE, et al., Eur J Clin Microbiol Infect Dis, 35:1069–1077, 2016, Chang HW, et al., Microbiome, 6:154-27, 2018). S. epidermidis also inhibits the growth of Cutibacterium acnes, which is associated with acne vulgaris (AV) (Wang Y., et al., Appl Microbiol Biotechnol, 98:411-424, 2014, Christensen GJM, et al., BMC Genomics, 17:152-14, 2016, Nakamura K., et al., Sci Rep, 10:21237-12, 2020). Therefore, S. epidermidis might play a key role in various skin disorders."

Page 5, lines 63-64.

“In this present study, we investigated the effect of D. acidovorans on the growth of S. epidermidis and elucidated its mechanisms.”

Page 5, lines 67-70.

“Our findings provide important insight into how D. acidovorans affects the skin microbiome by inhibiting the growth of S. epidermidis, resulting in a skin microbiome imbalance related to various skin disorders.”

Page 15-16, lines 227-237.

" In the skin of patients with AD, the relative abundance of S. aureus is significantly increased, and enterotoxin or protease derived from S. aureus exacerbates inflammation of AD. The relative abundance of S. aureus is also increased in the skin of patients with PS and Th17 cytokines are induced by S. aureus. S. epidermidis inhibits the growth of S. aureus, which might be associated with AD and PS. Moreover, S. epidermidis inhibits the growth of C. acnes, which is associated with AV. We therefore hypothesized that the inhibitory activity of substances secreted by D. acidovorans against S. epidermidis is indirectly associated with exacerbation of AD, PS, and AV. D. acidovorans might be related to various skin disorders. Additional studies are needed to elucidate relationship between D. acidovorans and AD, PS, and AV. "

Figure legends are within the ms-text (and sometimes in interrogative case. I would suggest to keep the legends narrative, and all legends together after references.

According to the reviewer’s suggestion, we changed the location of the figure legends in the revised manuscript.

RNA seq data should be available on some public database and be cited in ms accordingly.

According to the reviewer’s suggestion, we deposited our raw data of the RNA seq analysis with the DDBJ database (Submission ID = mpu_microbiology-0001, BioProject = PRJDB11464, BioSample = SAMD00294166, SAMD00294165, Accession number = DRA011812). Moreover, we added a data availability statement section describing this in the revised manuscript (page 17, lines 262-267).

6. PLOS authors have the option to publish the peer review history of their article (what does this mean?). If published, this will include your full peer review and any attached files.

We would like to publish the peer review history of this article.

---

## [Decision Letter · Decision Letter 1]

9 Jun 2021

Delftia acidovorans secretes substances that inhibit the growth of Staphylococcus epidermidis through TCA cycle-triggered ROS production

PONE-D-21-01227R1

Dear Dr. Sugita,

We’re pleased to inform you that your manuscript has been judged scientifically suitable for publication and will be formally accepted for publication once it meets all outstanding technical requirements.

Kind regards,

Rashid Nazir

Academic Editor

PLOS ONE

Additional Editor Comments (optional):

Reviewers' comments:

Reviewer's Responses to Questions

**Comments to the Author**

1. If the authors have adequately addressed your comments raised in a previous round of review and you feel that this manuscript is now acceptable for publication, you may indicate that here to bypass the “Comments to the Author” section, enter your conflict of interest statement in the “Confidential to Editor” section, and submit your "Accept" recommendation.

Reviewer #1: (No Response)

Reviewer #2: All comments have been addressed

2. Is the manuscript technically sound, and do the data support the conclusions?

Reviewer #1: Partly

Reviewer #2: Yes

3. Has the statistical analysis been performed appropriately and rigorously? 

Reviewer #1: N/A

Reviewer #2: Yes

4. Have the authors made all data underlying the findings in their manuscript fully available?

Reviewer #1: Yes

Reviewer #2: Yes

5. Is the manuscript presented in an intelligible fashion and written in standard English?

Reviewer #1: Yes

Reviewer #2: Yes

6. Review Comments to the Author

Reviewer #1: Dear authors,

Please change your abstract by placing more emphasis on the results of your work. As your work is not directly on atopic dermatits, omit the first line of the abstract.

Reviewer #2: The revision is satisfactory and I appreciate the responses. Furthermore, the data now has been deposited in public repositories that will help the readers for futuristic studies.

7. PLOS authors have the option to publish the peer review history of their article (what does this mean?). If published, this will include your full peer review and any attached files.

Reviewer #1: **Yes: **Azin Ayatollahi

Reviewer #2: No

---

## [Editor Report · Acceptance letter]

22 Jun 2021

PONE-D-21-01227R1 

*Delftia acidovorans* secretes substances that inhibit the growth of *Staphylococcus epidermidis* through TCA cycle-triggered ROS production. 

Dear Dr. Sugita:

I'm pleased to inform you that your manuscript has been deemed suitable for publication in PLOS ONE. Congratulations! Your manuscript is now with our production department. 

Kind regards, 

on behalf of

Dr Rashid Nazir 

Academic Editor

PLOS ONE